# Impact of Hot-Melt-Extrusion on Solid-State Properties of Pharmaceutical Polymers and Classification Using Hierarchical Cluster Analysis

**Ioannis Partheniadis [1][iD], Miltiadis Toskas [1], Filippos-Michail Stavras [1], Georgios Menexes [2][iD] and Ioannis Nikolakakis [1],*[iD]**

[1] Department of Pharmaceutical Technology, School of Pharmacy, Faculty of Health Sciences, Aristotle University of Thessaloniki, 54124 Thessaloniki, Greece; ioanpart@pharm.auth.gr (I.P.); toskasmi@pharm.auth.gr (M.T.); stavrasf@pharm.auth.gr (F.-M.S.)

[2] Laboratory of Agronomy, School of Agriculture, Aristotle University of Thessaloniki, 54124 Thessaloniki, Greece; gmenexes@agro.auth.gr

* Correspondence: yannikos@pharm.auth.gr; Tel.: +30-2310-997635

**Abstract:** The impact of hot-melt extrusion (HME) on the solid-state properties of four methacrylic (Eudragit® L100-55, Eudragit® EPO, Eudragit® RSPO, Eudragit® RLPO) and four polyvinyl (Kollidon® VA64, Kollicoat® IR, Kollidon® SR, and Soluplus®) polymers was studied. Overall, HME decreased Tg but increased electrostatic charge and surface free energy. Packing density decreased with electrostatic charge, whereas Carr's and Hausner indices showed a peak curve dependency. Overall, HME reduced work of compaction (Wc), deformability (expressed as Heckel $P_Y$ and Kawakita 1/b model parameters and as slope S' of derivative force/displacement curve), and tablet strength (TS) but increased elastic recovery (ER). TS showed a better correlation with S' than $P_Y$ and 1/b. Principal component analysis (PCA) organized the data of neat and extruded polymers into three principal components explaining 72.45% of the variance. The first included Wc, S' and TS with positive loadings expressing compaction, and ER with negative loading opposing compaction; the second included $P_Y$, 1/b, and surface free energy expressing interactivity with positive loadings opposing tap density or close packing. Hierarchical cluster analysis (HCA) assembled polymers of similar solid-state properties regardless of HME treatment into a major cluster with rescaled distance Cluster Combine Index (CCI) < 5 and several other weaker clusters. Polymers in the major cluster were: neat and extruded Eudragit® RSPO, Kollicoat® IR, Kollidon® SR, Soluplus®, and extruded Eudragit® L100-55. It is suggested that PCA may be used to distinguish variables having similar or dissimilar activity, whereas HCA can be used to cluster polymers based on solid-state properties and pick exchangeable ones (e.g., for sustain release or dissolution improvement) when the need arises.

**Keywords:** factor analysis; classification; dendrograms; hot-melt extrusion; polymers; powders

## 1. Introduction

Hot-melt extrusion (HME) is increasingly employed in the pharmaceutical industry for the production of solid dispersions of drugs with polymeric carriers. It is used for solubility enhancement [1,2], controlled, sustained and targeted delivery [3–9], and taste-masking [10,11]. Its efficiency is due to the high shear stresses exerted by the rotating screw in the barrel to molten drug/polymer mixture producing amorphous or fine crystalline solid drug dispersions [12]. As a manufacturing method, it has desirable characteristics, being continuous, solvent-free, and easily scaled-up [7,13,14].

For the application of HME, thermoplastic polymers are used based on stability over a wide temperature range, miscibility with the drug, and desired properties of the final dosage form. They are

used alone or with plasticizers which are added in small amounts to decrease Tg and melt viscosity. This, in turn, reduces processing time and temperature, thus minimizing potential risks on drug stability [15]. In many industrial applications, downstream processing follows, whereby the extruded solidified melt is milled into powder and compressed into tablets [4,12].

Several studies have reported adverse effects of HME on the mechanical properties of polymers. Grymonpré and co-workers [16,17] found that it increased the elastic recovery of PVA tablets and altered the compression mechanism of mixtures of the amorphous polymers Soluplus®, Kollidon® VA 64, Eudragit® EPO with Celecoxib towards fragmentation. As a result, HME may impact strain tolerance under compressive stress and this may promote cracking distress [18]. One way to ameliorate the effects of HME on tabletability is by compression at elevated temperatures [19], another way is by using plasticizers [3]. So far, there is no systematic study on the impact of HME on the entire spectrum of solid-state properties of extrudates, at the different processing stages from powder to the final tablet.

Therefore, the objective of this work was to study a range of solid-state properties of neat and extruded polymers, including particulate (a surface electrostatic charge, surface free energy, powder packing density, and packing indices), compression (work of compaction, elastic recovery, Heckel, Kawakita model parameters and slope of derivatized compression curve as indicators of plastic deformation) and tablet strength. Eight polymers with different pharmaceutical functionalities were studied as supplied powders and powdered extrudates obtained by milling and sieving. These were four methacrylic based polymers: Eudragit® RSPO and Eudragit® RLPO (extended-release), Eudragit® EPO (dissolution aid) and Eudragit® L100-55 (enteric release), and four polyvinyl based polymers: Kollidon® VA 64, Kollicoat® IR and Soluplus (dissolution aids) and Kollidon® SR (extended-release).

The results were subjected to two types of statistical analysis. ANOVA was applied first to evaluate the main effects and interactions of polymer and HME treatment. Next, Principal Component Analysis (PCA) was applied to extract groups of interrelated responses within each group [20] followed by hierarchical cluster analysis (HCA). The latter can be used as a confirmatory method to assemble polymers regardless of chemical nature and treatment into clusters of similar solid-state properties [21,22]. Therefore, HCA may be useful to select replacement polymers from the same cluster if needed, for example, due to shortage, financial cost, or incompatibility. Clusters are presented as dendrograms and cluster dissimilarity is expressed by the distance cluster combine index (CCI).

## 2. Materials and Methods

### 2.1. Materials

Eudragit® L100-55 (EL), Eudragit® EPO (EE, after milling/sieving the supplied pellets), Eudragit® RSPO (RS), and Eudragit® RLPO (RL, after milling/sieving) were gifted by Evonik (Darmstadt, Germany) and Kollidon® VA 64 (VA), Kollicoat® IR, (IR), Kollidon® SR (SR), Soluplus® (SL) gifted by BASF (Ludwigshafen, Germany). Extruded polymers are denoted with subscript (ex). Lutrol® F 127 (polyoxyethylene with polyoxypropylene copolymer) gift by BASF was used as a plasticizer for Eudragit® L 100-55 only.

### 2.2. Methods

#### 2.2.1. Hot-Melt Extrusion (HME)

Codes of polymer samples, conditions during hot-melt extrusion, and polymer decomposition temperatures are given in Table 1. A bench-type vertical single-screw extruder (Model RCP-0250 Microtruder, Randcastle Extrusion Systems, NJ, USA) fitted with a 2 mm orifice die was used at 40 rpm screw speed. The zone temperature ranges varied from 80–120 °C to 150–210 °C and the barrel pressures from 0.41–0.45 to 8.63–13.38 MPa depending on the polymer (Table 1). For Eudragit® L100-55, 10% Lutrol® F 127 was added to enable extrusion. Extruded rods were cut into cylindrical pellets (RCP-0250 Micro Pelletizing System, Randcastle, NJ, USA) and then milled with a cutter mill (IKA A11, Koenigswinter, Germany).

The resulting powder was sieved (Retch Analytical sieves 425, 150, 106, and 75 µm tied to Fritsch Analysette 3 PRO Vibratory Platform) and the fraction as close as possible to the corresponding neat powder was selected for the experiments. Experimental powders were kept in closed glass containers.

**Table 1.** Polymer, coding, and conditions during hot-melt extrusion and polymer decomposition temperatures.

| Polymer | Processing | Code | Temperature Zones (°C) | | | | Tdec (°C) | Barrel Pressure (MPa) |
|---|---|---|---|---|---|---|---|---|
| | | | T1 | T2 | T3 | T4 | | |
| Eudragit® L100-55 | Supplied Extruded/milled | EL ELex | 90 | 125 | 140 | 140 | 174.2 | 9.99–11.03 |
| Eudragit® EPO | Supplied/milled Extruded/milled | EE EEex | 80 | 105 | 115 | 120 | 223.6 | 6.55–7.24 |
| Eudragit® RSPO | Supplied Extruded/milled | RS RSex | 80 | 110 | 120 | 125 | 162.6 | 8.62–8.96 |
| Eudragit® RLPO | Supplied/milled Extruded/milled | RL RLex | 85 | 110 | 120 | 125 | 160.3 | 8.63–11.38 |
| Kollidon® VA64 | Supplied Extruded/milled | VA VAex | 135 | 160 | 170 | 175 | 246.2 | 0.28–0.48 |
| Kollicoat® IR | Supplied Extruded/milled | IR IRex | 150 | 200 | 210 | 210 | 239.9 | 0.41–0.45 |
| Kollidon® SR | Supplied Extruded/milled | SR SRex | 90 | 125 | 135 | 145 | 238.1 | 2.07–2.41 |
| Soluplus® | Supplied Extruded/milled | SL SLex | 110 | 170 | 175 | 180 | 248.9 | 0.66–0.72 |

### 2.2.2. Particle Size, Shape, and Density

Particle size and shape of neat and extrudate powder were estimated using optical microscopy and image analysis system comprised of Olympus BX41 microscope fitted with U-SPT and U-PMTVC extensions (Tokyo, Japan), Leica DF295 video camera (Wetzlar. Germany), and Leica Microsystems software (Heerbrugg, Switzerland) by analyzing about 300 particles. Particle size was expressed as mean equivalent circle diameter and shape as roundness index [= perimeter$^2$/(12.56 × mean projection area)] with value one for the sphere, increasing with shape irregularity. Particle density ($\rho_s$) was determined with helium pycnometry (Ultrapycnometer 1000, Quantachrome Instruments, Boynton, FL, USA), from the volume of accurately weighted samples after calibration with a standard 7.0699 cm$^3$ steel ball.

### 2.2.3. Differential Scanning Calorimetry (DSC)

Thermal analysis was applied to identify thermal changes of the polymers due to HME processing using differential scanning calorimetry (DSC-50, Shimadzu Corporation, Kyoto, Japan connected to TA-60 WS data acquisition unit). Samples of about 7 mg weight were placed in aluminum pans, sealed and heated in the range 25–200 at 10 °C/min under a nitrogen gas flow of 50 mL/min. The calorimeter was calibrated using indium standards. When glass transition appeared as peak, this was taken as Tg and when the transition appeared as baseline shift, the mid-point of the deflection was used. DSC measurements were taken 2–3 days after extrusion.

### 2.2.4. Electrostatic Surface Charge

The electrostatic surface charge of the particles was measured with a resolution of 1 pC using a Faraday pail (JCI 140, Chilworth Technology Ltd., Southampton, UK) connected to a monitor screen for visual display (JCI 147, Chilworth Technology Ltd., Southampton, UK). About 1 g polymer powder was dropped into the center of the Faraday pail and the charge was read directly in nC.

### 2.2.5. Surface Free Energy

The surface free energy $\gamma_s$ and energy components of the powders were obtained from contact angle measurements using distilled water and methylene iodide (analytical grade, Aldrich, Taufkirchen, Germany) as test liquids and a horizontal traveling microscope for measuring the drop height. The liquid was added dropwise to the surface of compressed powder (3.0 × 27.0 mm, of total porosity 0.10–0.25) and height was continuously recorded with 0.3 μm accuracy using an LVDT transducer attached to the microscope and connected to a signal conditioner (E309, RDP Electronics, Wolverhampton, UK) and data acquisition unit (Handyscope TiePie Electronics, Sneek, The Netherlands) [23,24].

### 2.2.6. Powder Packing

The bulk ($\rho_b$) and tap ($\rho_t$) densities were determined with a USP1 tester at 14 mm vertical drop, (model SVM 101, Erweka Heusenstamm, Germany) fitted with a 25 mL volumetric cylinder. $\rho_b$ was measured from the volume reading after pouring known powder weight into the cylinder via a funnel and $\rho_t$ from the reading after 300-cylinder taps. From the $\rho_b$ and $\rho_t$ values Hausner's ratio HR (= $\rho_b/\rho_t$) [25] and Carr's index CC% (= ($\rho_b - \rho_t$)/$\rho_t$) [26] were determined as indices of packing ability.

### 2.2.7. Powder Compression

Tablets (100 mg) were prepared using an instrumented press (Ø6.0 mm flat-edged punches, Gamlen D-Series Press, Nottingham, UK) at 173 MPa compression pressure and 10 mm/min speed. Force-displacement curves were recorded during compression/decompression (Figure 1). From these, the work of compaction ($Wc$), Equation (1) was obtained by subtracting the total area ($0F'X'$) defined by the beginning of compression ($O$) to the peak force ($F'$) and point $X'$ of the line drawn from $F'$ perpendicular to the displacement axis, minus the decompression area ($XF'X'$). Elastic recovery (ER) (Equation (2)) was calculated as the %difference of compact thickness at maximum punch penetration ($X'$) and at ejection ($X$) relative to $X'$.

$$Wc = \int_0^x dx - \int_x^{x'} dx \qquad (1)$$

$$ER\,\% = \frac{x' - x}{x} \qquad (2)$$

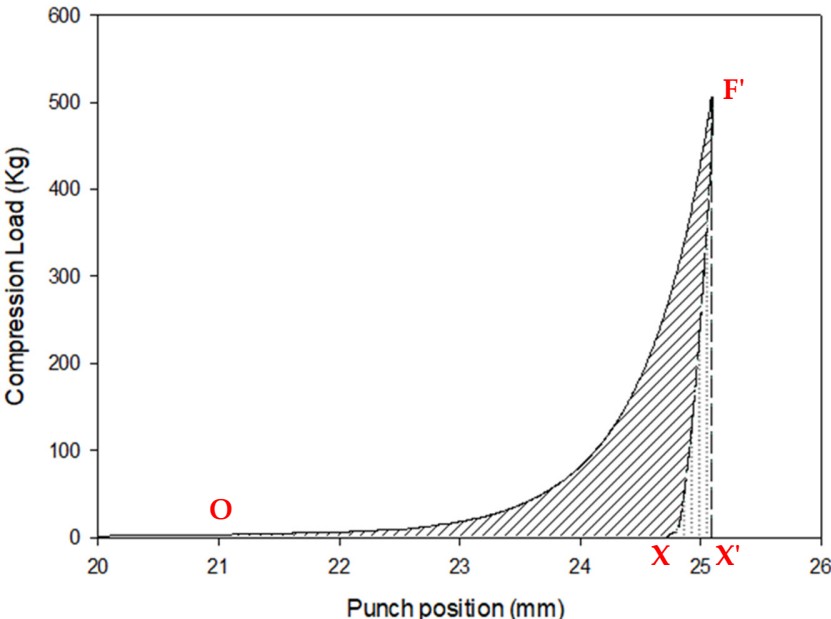

**Figure 1.** Representative force—displacement profiles recorded during compression and decompression.

Additionally, derivative compression pots of *dX/dF* (punch displacement over compression force change), against 1/F′ were constructed and the slopes *S*′ of the resulting straight lines (Equation (3)), were estimated as deformability index [27].

$$\frac{dx}{dF} = \frac{S'}{F} + constant \tag{3}$$

### 2.2.8. Compression Models

Heckel model is expressed by Equation (4) in which the reciprocal slope 1/*k* is the yield pressure ($P_Y$) [28]:

$$\ln\left[\frac{1}{1 - p_F}\right] = A + kP \tag{4}$$

*P* is the compression pressure, $p_F$ the solid fraction [= compact weight/(volume × particle density)]. Constant *A* is related to volume reduction due to die filling and rearrangement before compaction begins. Since Heckel plots consist of three sequential regions of initial packing, plastic deformation, and elastic/hardening [29–31] piece-wise regression was applied to define the intermediate region of plastic deformation and obtain 1/*k* [32].

Kawakita model is expressed by Equation (5) where *C* is the degree of volume reduction (Equation (6)) [33].

$$\frac{P}{C} = \frac{P}{a} + \frac{1}{a \times b} \tag{5}$$

$$C = \frac{V_0 - V}{V_0} \tag{6}$$

$V_0$, *V* are the initial volume and at pressure *P*, and *a*, *b* constants obtained from *P/C* vs. *P* linear plot. The parameter 1/*b* is related to the yield strength of particles [34].

### 2.2.9. Tablet Strength

Tensile strength (TS) was measured by diametrical loading, using the instrumented press described previously for compression but operated in fracture mode, fitted with a 10 kg load cell according to the United States Pharmacopoeia monograph 1217 [35]. TS was obtained from Equation (7) [36]:

$$TS = \frac{2L}{4\pi\Phi h} \tag{7}$$

where *L* is the breaking load, $\Phi$ the tablet diameter, and *h* its thickness.

### 2.2.10. Experimental Design and Statistical Analysis

A factorial design was employed with two factors: polymer at eight and treatment (neat or extruded powder) at two levels, making up a total of 16 experimental combinations. Experiments were performed in triplicate. For testing the main effects and interactions ANOVA was applied. Least Significant Difference (LSD) between means of a certain variable was also calculated (Equation (8)).

$$LSD = t_{0.025} \sqrt{\frac{2MS}{n}} \tag{8}$$

$t_{0.025}$ is read from t-distribution tables, *MS* is the mean square error obtained from ANOVA and *n* number of repetitions. Pearson's *r* correlation coefficient was used to identify correlations between variables.

### 2.2.11. Principal Component (PCA) and Hierarchical Cluster Analysis (HCA)

PCA explores relationships and defines groups of correlated response variables. These groups represent dimensions (components) within the data. Loadings in a component represent correlations

of each variable with the respective component. KMO index and Bartlett's test of sphericity were used as criteria for reliable analysis. Components with Eigenvalue > 1 and variables with loadings > 0.50 were examined. VARIMAX rotation was run to obtain component structure and variable loadings [37]. Prior to PCA, data were checked for outliers and the linear relationships among the tested variables were examined by inspecting the corresponding scatter plots. The quality of the data was confirmed from the satisfactory value of the KMO index of sampling adequacy ($\approx$0.60 > 0.50) and the statistical significance of Bartlett's test of sphericity ($p < 0.001$).

HCA was used to identify clusters of polymers with similar solid-state properties, regardless of HME treatment. HCA clusters cases within the data although it does not validate their existence [37]. For this reason, it was applied to components derived from the PCA. It begins by assuming that each case (polymer) forms its own cluster which is joined progressively by similar clusters until only one remains. An important feature of HCA is that results obtained at an earlier stage are set within results obtained later, thus creating a tree-structure (dendrogram). For the evaluation of HCA, the single-linkage joining method algorithm (nearest-neighbor method) was used. It is versatile, able to reveal different patterns, and defines cluster similarity as the shortest Euclidean distance [37]. All statistical analyses (ANOVA, PCA, HCA) were conducted with SPSS 20.0 (IBM Statistics, Inc., Chicago, IL, USA).

## 3. Results and Discussion

### 3.1. Hot-Melt Extrusion (HME)

From Table 1 it appears that lower extrusion temperatures were needed for the four methacrylic polymers EL (90 to 140 °C), EE (80 to 120 °C) RS, and RL (80 to 125 °C) and for SR (90 to 145 °C) compared to VA (135 to 170 °C), IR (150 to 210 °C) and SL (110 to 175 °C). Comparing the extrusion temperatures with the Tg values of the polymers in Table 2, it appears that for the methacrylics the first zone temperature (T1) was about 20 to 25 °C higher than the respective Tg and for VA and SR about 40 °C, which is within the expected range [38,39]. However, for polymer IR quite high temperatures had to be applied, with T1 exceeding Tg by 100 °C (Tables 1 and 2). This is ascribed to IR crystallinity as explained in Section 3.3, requiring higher energy to break intra- and inter-chain bonds between the ordered polymer chains, than the energy to disentangle chains of amorphous polymers and initiate mobility [13]. In all cases, the applied temperatures were well below decomposition (Tdec, Table 1). Furthermore, from Table 1 it appears that methacrylic polymers developed higher barrel pressures than polyvinyl polymers (6.55–7.24 to 8.63–11.38 MPa compared with 0.41–0.45 to 2.07–2.41) indicating higher resistance to extrusion or higher melt viscosity.

**Table 2.** Particle density, size, shape, and glass transition temperature of the experimental polymeric powders.

| Polymer Code | $\rho_s$ (g/cc) | $d_{50}$ (μm) | Shape Index | Tg (°C) |
|---|---|---|---|---|
| EL | 1.301 ± 0.01 | 26 | 1.17 | 69.9 ± 0.1 |
| ELex | 1.299 ± 0.01 | 54 | 1.76 | 61.6 ± 0.1 |
| EE | 1.396 ± 0.01 | 76 | 1.68 | 56.7 ± 0.2 |
| EEex | 1.372 ± 0.02 | 64 | 1.81 | 54.9 ± 0.1 |
| RS | 1.380 ± 0.01 | 30 | 1.16 | 60.8 ± 0.1 |
| RSex | 1.361 ± 0.01 | 62 | 1.95 | 55.5 ± 0.2 |
| RL | 1.345 ± 0.01 | 38 | 1.79 | 60.5 ± 0.1 |
| RLex | 1.332 ± 0.01 | 57 | 1.94 | 54.2 ± 0.1 |
| VA | 1.219 ± 0.01 | 22 | 1.90 | 97.2 ± 0.3 |
| VAex | 1.217 ± 0.01 | 68 | 1.92 | 86.3 ± 0.1 |
| IR | 1.272 ± 0.01 | 47 | 1.42 | 50.9 ± 0.2 |
| IRex | 1.254 ± 0.01 | 71 | 1.74 | 45.6 ± 0.3 |
| SR | 1.331 ± 0.02 | 62 | 1.35 | 45.5 ± 0.1 |
| SRex | 1.302 ± 0.01 | 70 | 1.79 | 42.3 ± 0.2 |
| SL | 1.371 ± 0.02 | 71 | 1.21 | 68.9 ± 0.2 |
| SLex | 1.323 ± 0.01 | 62 | 1.68 | 59.4 ± 0.2 |

$\rho_s$: particle density (mean ± standard deviation (SD)); $d_{50}$ median particle diameter; Tg glass transition temperature (mean ± standard deviation).

### 3.2. HME and Particle Density, Size and Shape

In Table 2, the results of particle density, size, and shape are presented. The densities of neat and extruded polymers were similar (from 1.219 to 1.396 g/cc and 1.202 to 1.372 g/cc, respectively). The largest difference (from 1.371 to 1.323 g/cc) is seen for SL and is associated with its large Tg decrease after extrusion (from 68.9 to 59.4 °C, Table 2), signifying greater structural changes and open-chain arrangement. Furthermore, from Table 2 it is seen that polymers EE, RL, SR, and SL had similar d50 values as neat or powder extrudates. Conversely, EL, RS, VA, and IR that were supplied as spray-dried fine powders, had higher d50 as powder extrudates which is ascribed to the sticking of the milled extrudate particles to the sieve frames inhibiting usage of smaller aperture sieves.

Additional information on the morphology of the particles is provided by the SEM images in Figure 2. Neat EL and RS had spherical particles (shape indices 1.17 and 1.16, Table 2). VA was a mixture of spherical and broken particles (shape index 1.90), and SR, SL had roundish particles (shape index 1.35 and 1.21, Table 2), whereas extrudate powders had angular particles (shape index between 1.68 and 1.92) due to the shearing action of the cutter mill. The ANOVA (Table 3) showed a significant interaction of the effects of polymer and HME/milling on particle shape. This is explained from the images in Figure 2 where polymers Eudragit® EPO, Eudragit® RLPO had irregularly-shaped particles both as neat and extruded/milled powders, whereas Eudragit® L100–55, Eudragit® RSPO and Soluplus® had spherical particles as neat powders but irregular as extruded/milled.

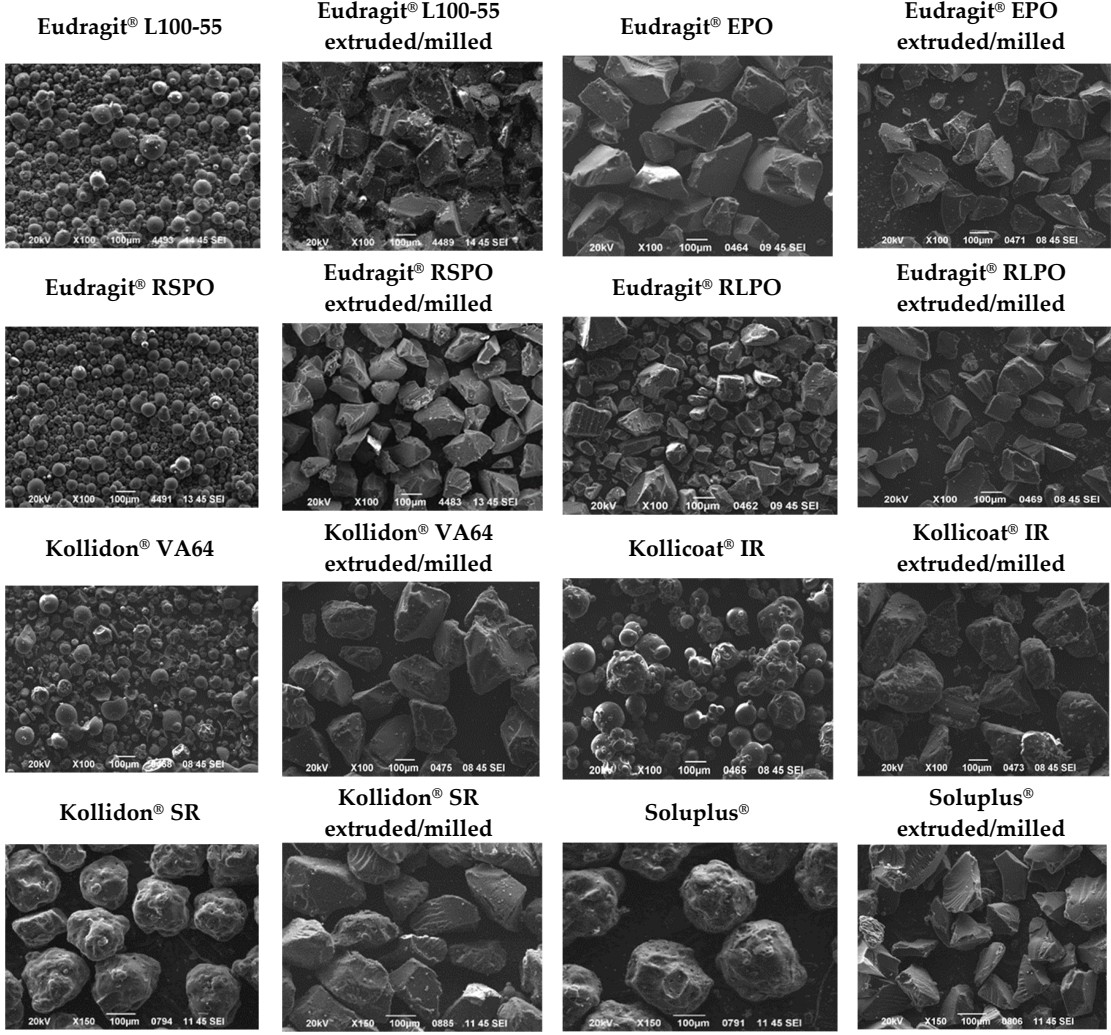

**Figure 2.** SEM microphotographs of the particles of neat and extruded experimental polymeric powders.

### 3.3. HME and Thermal Properties

Results of the thermal analysis are presented in Figure 3 separately for methacrylic and polyvinyl experimental polymers and Tg values are given in Table 2. As apart from HME processing polymers had the same thermal history, DSC scans of the first heating cycle where the events were more discernible, were analyzed. Polymers were amorphous except IR which was semi-crystalline, melting at 209.5 °C the neat polymer, and at higher 213.7 °C temperature the extruded form (insert in Figure 3b). The increase is because of the increase in the crystallinity of the PVA fraction of the polymer due to HME reported earlier [40]. In the scans of SR and SL, glass transitions appear as small or shallow endothermic peaks, whereas in the case of VA as a step. The melting peak at 51.6 °C in the EL scan is due to the presence of Lutrol 127. This peak has merged with the glass transition event in the scan of the extruded polymer indicating single phase formation [41].

From Table 2 it appears that VA has the highest Tg as neat or extruded (97.2 °C /86.3 °C) followed by SL (68.9 °C/59.4 °C) and EL (69.9 °C/61.6 °C). These three polymers also needed higher extrusion temperatures (Table 1). From Table 2 it appears that Tg always decreased after HME. ANOVA results of the effects of polymer and HME on Tg are presented in Table 3 and show significant interaction of the effects of polymer and HME treatment. In particular, the decrease in Tg is greater for polymers EL, VA, SL (by 8.3, 10.9, 9.5 °C) with high neat polymer Tg or more rigid structure, and lower for EE and SR (by 1.8 and 3.2 °C) with lower Tg (Table 1). [42]. The decrease of Tg after HME is in agreement with previously published results and signifies restructuring of polymer chains to a more relaxed state [19,43].

**Table 3.** ANOVA results (statistical significance, p-values) of main effects and interactions of the polymer (A) and treatment (hot-melt extrusion (HME)) (B) on the properties of experimental polymers.

| Property | $R_a{}^2$ | Main Effects | | Interactions |
|---|---|---|---|---|
| | | **A** | **B** | **A × B** |
| Mean particle diameter (d50) | 0.999 | <0.001 | <0.001 | <0.001 |
| Shape index | 0.998 | | | |
| Particle density ($\rho_s$) | 0.949 | <0.001 | <0.001 | 0.031 |
| Glass transition temperature (Tg) | 0.999 | <0.001 | <0.001 | <0.001 |
| Surface charge (SC) | 0.985 | | | |
| Bulk density ($\rho_b$) | 0.984 | | | |
| Tapped density ($\rho_t$) | 0.990 | | | |
| Hausner's ratio (HR) | 0.962 | | | |
| Carr's compressibility index (CC) | 0.995 | | | |
| Dispersion component of free energy ($\gamma_s{}^d$) | 0.998 | | | |
| Polar component of free energy ($\gamma_s{}^P$) | 0.999 | | | |
| Surface free energy ($\gamma_s$) | 0.997 | <0.001 | 0.081 | <0.001 |
| Work of compaction (Wc) | 0.920 | <0.001 | <0.001 | <0.001 |
| Elastic recovery (ER) | 0.979 | | | |
| Heckel's yield pressure ($P_Y$) | 0.994 | | | |
| Kawakita parameter "1/b" | 0.959 | | | |
| Slope of derivatized compression plots | 0.967 | | | |
| Tensile strength (TS) | 0.996 | | | |

### 3.4. HME and Electrostatic Charge

The presence of electrostatics has been found to deteriorate flowability and tableting performance [44,45]. Since most of the present experimental materials were ionic or carried potentially ionizable groups the effect of HME on electrostatics was examined. The results of electrostatic charge measurements of the experimental powders are presented in Table 4. Overall, the measured values were small because measurements were taken on powders equilibrated for several days after HME processing without any

triboelectrification applied prior to experiments. Methacrylic polymers EE, RS, and RL with quaternary cationic ammonium groups displayed positive surface charges (2.29, 1.02, 1.04 nC). EL is also positively charged (1.18 nC) despite the presence of negatively charged carboxyl groups in its structure that may be ascribed to the adsorption of counterions from contacting surfaces during packaging and transportation. The positive charge on the polyvinyl based polymers VA, SR, and SL can be attributed to the nitrogen in the lactam ring.

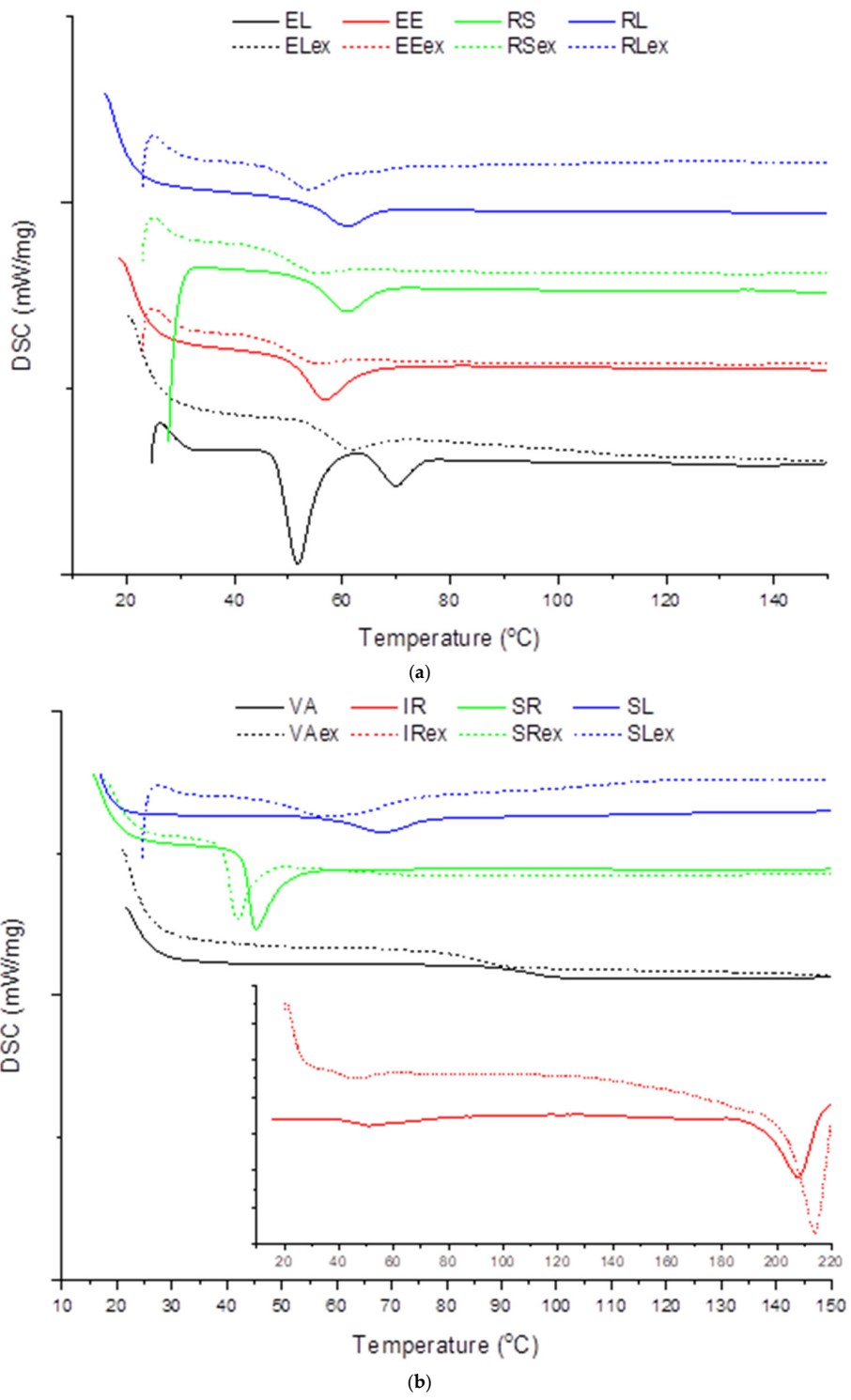

**Figure 3.** Differential scanning calorimetry (DSC) thermographs of neat and extruded methacrylic (**a**) and polyvinyl (**b**) polymers.

From Table 4 it can be seen that HME changed significantly the electrostatic charge of extrudate powders (differences > $LSD_{0.05}$ = 0.09 nC). Polymer IR with no chargeable groups showed the lowest increase (0.27/0.40, neat/extruded). Methacrylic polymers EE, RS, RL, and polyvinyl VA, SR demonstrated low to medium increases (0.78, 2.74, 1.19, 0.34, and 1.36 nC, respectively) whereas SL showed a large decrease (4.53 nC) possibly associated with alteration of acetic acid/acetate ratio at the particle surface [39]. The negative charge of ELex is ascribed to the presence of Lutrol® F127 [42]. The measured charge on Lutrol® F127 powder alone was −4.78 nC. The increase of electrostatic charge of powder extrudates can be explained by the intense shearing of polymer melt during HME causing exposure of existing charged groups to a greater extend.

### 3.5. HME and Surface Free Energy

In Table 4 results of the dispersive ($\gamma_s^d$) and polar ($\gamma_s^p$) components and of the total surface free energy ($\gamma_s$) obtained from contact angle measurements are presented for neat and extrudate powders. The $\gamma_s$ increased for EL, EE, VA, and SR, but for RL and IR decreased, whereas for RS, SL the change was very small. The $\gamma_s^d$ increased for EL, EE, VA, and SR, but for RL and IR decreased, whereas for RS and SL the change was small. The $\gamma_s^p$ increased for RS, RP, VA, IR, and SL, but decreased for EL, EE, and SR. Except for EE, the changes in $\gamma_s^d$ and $\gamma_s^p$, led to a significant increase of $\gamma_s$. The decrease for EL is ascribed to the presence of the surface-active Lutrol 127 in the extruded polymer [46]. The increase of $\gamma_s$ is large for RS (from 48.0 to 69.9 mN/m), RL (49.8 to 65.9 mN/m) and VA (49.2 to 60.5 mN/m), intermediate for SR (60.8 to 65.9 mN/m) and SL (47.6 to 53.4 mN/m) and small for EE (60.8 to 61.1 mN/m). The different responses of the polymers' surface free energy to HME is seen in the ANOVA (Table 3) as significant interaction. The increased $\gamma_s$ of extrudate powders may alter their behavior during mixing, flow, wetting, and dissolution [24,47–49].

Considering the effects of HME on $\gamma_s$ in parallel with Tg (Tables 2 and 4), it appears that polymer EE, with a small decrease in Tg (1.8 °C) showed a small $\gamma_s$ increase (0.31 mN/m), whereas VA with a large decrease in Tg (10.9 mN/m) showed large $\gamma_s$ increase (11.3 mN/m). However, this trend is not general since polymers RS, IR, with the same Tg change (5.3 mN/m) demonstrated different $\gamma_s$ change after HME, the former high increase (21.9 mN/m) but the latter low increase (3.73 mN/m). Thus, the overall structural changes resulting from the intense shearing and heating during HME, as expressed by Tg change, do not always reflect on surface chemistry [44,50].

**Table 4.** Results of electrostatic surface charge and surface free energy components (mean values ± SD, *n* = 3, Least Significant Difference $LSD_{0.025}$).

| Polymer Code | Electrostatic Surface Charge (nC) | $\gamma_s^d$ (mN/m) | $\gamma_s^p$ (mN/m) | $\gamma_s$ (mN/m) |
|---|---|---|---|---|
| EL | 1.18 ± 0.21 | 26.3 ± 0.12 | 40.0 ± 0.14 | 66.3 ± 0.26 |
| ELex | −0.98 ± 0.09 | 28.6 ± 0.24 | 34.2 ± 0.19 | 62.7 ± 0.43 |
| EE | 2.29 ± 0.18 | 22.8 ± 0.08 | 38.0 ± 0.24 | 60.8 ± 0.32 |
| EEex | 3.07 ± 0.11 | 30.1 ± 0.19 | 30.9 ± 0.10 | 61.1 ± 0.29 |
| RS | 1.02 ± 0.03 | 31.2 ± 0.11 | 16.8 ± 0.24 | 48.0 ± 0.35 |
| RSex | 3.03 ± 0.31 | 31.6 ± 0.15 | 38.3 ± 0.27 | 69.9 ± 0.42 |
| RL | 1.04 ± 0.12 | 32.3 ± 0.24 | 17.5 ± 0.29 | 49.8 ± 0.53 |
| RLex | 2.23 ± 0.17 | 29.5 ± 0.29 | 36.4 ± 0.21 | 65.9 ± 0.50 |
| VA | 1.44 ± 0.23 | 16.0 ± 0.18 | 33.3 ± 0.25 | 49.2 ± 0.43 |
| VAex | 1.78 ± 0.31 | 22.5 ± 0.14 | 38.0 ± 0.18 | 60.5 ± 0.32 |
| IR | 0.27 ± 0.04 | 24.1 ± 0.04 | 30.4 ± 0.14 | 54.5 ± 0.18 |
| IRex | 0.40 ± 0.01 | 23.8 ± 0.26 | 34.5 ± 0.09 | 58.3 ± 0.35 |
| SR | 0.57 ± 0.04 | 19.0 ± 0.36 | 41.8 ± 0.35 | 60.8 ± 0.71 |
| SRex | 1.93 ± 0.25 | 27.2 ± 0.12 | 38.7 ± 0.27 | 65.9 ± 0.39 |
| SL | 7.16 ± 0.47 | 25.5 ± 0.05 | 22.0 ± 0.24 | 47.6 ± 0.29 |
| SLex | 2.63 ± 0.24 | 25.1 ± 0.16 | 28.4 ± 0.13 | 53.4 ± 0.29 |
| $LSD_{0.05}$ | 0.09 | 0.08 | 0.09 | 0.17 |

$\gamma_s^d$: dispersion component of surface free energy; $\gamma_s^p$: polar component of surface free energy; $\gamma_s$: surface free energy.

### 3.6. HME and Powder Packing

#### 3.6.1. Packing Densities

As changes in particle surface properties affect the bulk powder behavior and as HME was shown to alter them, the packing of neat and extrudate powders was compared. The results are presented in Table 5. Bulk densities ($\rho_b$) of neat and extrudate powders were similar ranging from 0.42 to 0.69 g/cc (EL, SL), and 0.42 to 0.63 (ELex, IRex). This similarity is because $p_b$ arises from powder deposition and the fill volume is affected mainly by the material density and entrapped air. On the other hand, tap densities ($p_t$) of neat powders were significantly greater (from 0.51 to 0.84 g/cc, (EL, SR)) than extrudate (from 0.47 to 0.74 g/cc (ELex, SRex)), and the differences between corresponding neat and extrudate powders were significant ($>LSD_{0.05} = 0.01$). As $p_t$ develops from many cylinder impacts causing powder bed dilations and particle rearrangements the differences should arise from varying particle-particle interactions. ANOVA (Table 3) showed statistical interaction, i.e., the effect of HME depends on the polymer.

In the case of RS and VA, the lower $p_t$ is attributed to its more than twice greater particle size (Table 2). However, for EE, RL, IR, SR, and SL with particles in a relatively narrow size range (38–76 μm), the decrease should be due to differences in the magnitude of interparticle surface forces which for a packed, non-compacted powder are long-range electrostatic [45,51,52].

**Table 5.** Results of powder packing properties (mean values ± SD, $n = 3$, $LSD_{0.025}$).

| Polymer | $\rho_b$ (g/cc) | $\rho_t$ (g/cc) | HR (%) | CC (%) |
|---|---|---|---|---|
| EL | 0.42 ± 0.01 | 0.51 ± 0.01 | 1.21 ± 0.03 | 17.7 ± 0.27 |
| ELex | 0.42 ± 0.01 | 0.47 ± 0.01 | 1.12 ± 0.01 | 10.6 ± 0.32 |
| EE | 0.53 ± 0.01 | 0.64 ± 0.01 | 1.21 ± 0.03 | 17.2 ± 0.38 |
| EEex | 0.53 ± 0.01 | 0.59 ± 0.01 | 1.11 ± 0.01 | 10.2 ± 0.08 |
| RS | 0.51 ± 0.01 | 0.65 ± 0.01 | 1.27 ± 0.02 | 21.5 ± 0.78 |
| RSex | 0.53 ± 0.01 | 0.59 ± 0.01 | 1.11 ± 0.01 | 10.2 ± 0.25 |
| RL | 0.56 ± 0.01 | 0.72 ± 0.01 | 1.29 ± 0.01 | 22.2 ± 0.24 |
| RLex | 0.54 ± 0.01 | 0.61 ± 0.01 | 1.13 ± 0.02 | 11.5 ± 0.19 |
| VA | 0.59 ± 0.01 | 0.78 ± 0.01 | 1.32 ± 0.01 | 24.4 ± 0.84 |
| VAex | 0.51 ± 0.01 | 0.66 ± 0.01 | 1.29 ± 0.03 | 22.7 ± 0.39 |
| IR | 0.65 ± 0.01 | 0.77 ± 0.01 | 1.18 ± 0.01 | 15.6 ± 0.24 |
| IRex | 0.63 ± 0.01 | 0.69 ± 0.01 | 1.10 ± 0.02 | 8.7 ± 0.17 |
| SR | 0.67 ± 0.01 | 0.84 ± 0.01 | 1.25 ± 0.01 | 20.4 ± 0.34 |
| SRex | 0.61 ± 0.01 | 0.74 ± 0.01 | 1.21 ± 0.01 | 17.6 ± 0.04 |
| SL | 0.69 ± 0.01 | 0.75 ± 0.01 | 1.09 ± 0.01 | 8.0 ± 0.48 |
| SLex | 0.60 ± 0.01 | 0.66 ± 0.01 | 1.10 ± 0.01 | 9.1 ± 0.24 |
| $LSD_{0.05}$ | 0.01 | 0.01 | 0.01 | 0.16 |

$\rho_b$: bulk density; $\rho_t$: tapped density; HR: Hausner's ratio; CC: Carr's compressibility index.

#### 3.6.2. Packing Indices

The differences in tap densities of neat and extrudate powders discussed previously are expected to affect the packing indices. From Table 5 it can be seen that except for SL, CC% is lower for the extrudate powders and ANOVA showed statistically significant polymer and HME interactions (Table 3). In particular, the decrease of CC% due to HME is greater for EL (7.1%), EE (7.0%), RS (11.4%), RL (10.7%) that also exhibited higher electrostatics change (2.16, 0.78, 2.74, 1.19 nC, Table 4), and smaller for VA (1.6%) which showed small increase (0.34 nC). The significant CC% reduction of IR (6.9%) despite a small charge increase (0.13 nC), is ascribed to the greater particle size of extrudate powder (71 μm compared to 47 μm). For SR which contrary to the other polymers exhibited a decrease of charge after HME, CC% showed a small increase (1.1%).

As there is no literature data on the dependency of packing indices on electrostatics, it was considered worthy to explore their relationship. This is tested in Figure 4 by plotting CC% (a) and HR

(b) against electrostatic charge using data of neat and extrudate powders (except SL which carried a very high charge, Table 4). It can be seen that polymers with low or high charge had low CC% and HR values, but those with intermediate charge (0.6 to 1.6 nC) had high indices. The low CC% and HR at low charge are explained by the easier packing due to minimal interaction, and the low values at high charge due to repulsion inhibiting close packing to smaller volumes. The high CC% and HR for powders with intermediate charge signifies the absorption of mechanical energy during tapping, leading to large volume reduction. Alteration of the packing of polymeric powders due to the effect of HME on electrostatics should be a factor to consider during capsule filling and tableting as it may affect capsule fill or die fill weight.

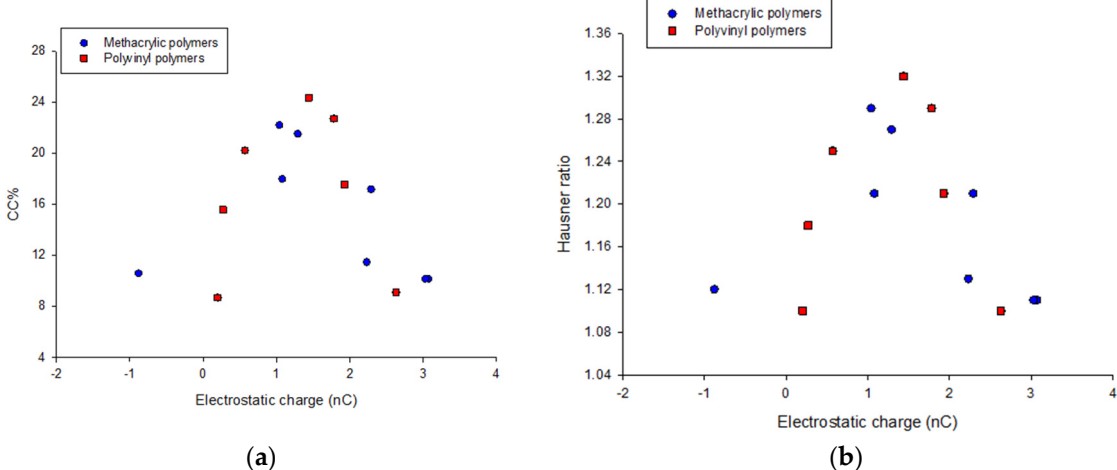

**Figure 4.** Plots of Carr's index (**a**) and Hausner ratio (**b**) against an electrostatic surface charge for neat/extruded methacrylic (circles) and polyvinyl based polymers (squares).

### 3.7. HME and Compression

Although there is published work on the impact of HME on the mechanical properties of polymers, few parameters have been used for the comparisons [3,16,19]. In the present work the effect of HME on the work of compaction (Wc), elastic recovery (ER), and slope (S') of derivative compression curves (dx/dF vs. 1/F') [27] was examined, besides Heckel and Kawakita model parameters. Wc, ER, and S' are directly obtained from 'in-die' force-displacement data, they are sensitive to materials and formulation, and hence useful for 'in-process' control according to the principles of process analytical technology (PAT) [19,53]. The values of compression parameters are given in Table 6.

### 3.7.1. Work of Compaction (Wc) and Elastic Recovery (ER)

The work of compaction is related to the absorbed energy. From Table 6 it can be seen that polymers EL, RS, VA, and IR showed significant Wc decrease (>$LSD_{0.05}$ = 0.11) which is attributed to the increased particle size of extrudate powder (Table 2) or less interparticle contact area. Polymers EE, RL, and SL with similar d50 before and after extrusion exhibited small changes. The significant decrease of Wc for SR, despite the similar neat/extrudate d50 should be ascribed to loss of deformability as indicated by the increase in Kawakita yield pressure parameter 1/b (from 25.3 to 39.3 MPa). Loss of deformability should also contribute besides particle size increase for the reduction of Wc of VA (by 1.8 MPa) and IR (by 0.88 MPa) as they both exhibited a large increase of 1/b (from 27.3 to 82.1 MPa and from 15.9 to 43.2 MPa). ANOVA showed a significant interaction of the effects of polymer and HME on Wc.

Elastic recovery controls the number of interparticle bonds remaining after the removal of the compression force. From Table 6 it appears that ER always increased after HME signifying changes in the internal material structure. A statistically significant increase (>$LSD_{0.05}$ = 0.36) is seen for EL,

RS, VA, IR, and SL whereas for EE, RL, SR the increase was not significant. The different response of polymers' ER to HME was confirmed by ANOVA (Table 3) as the interaction of the effects of polymer and HME. The increase of the ER of extrudate powders can be ascribed to structural relaxation and increased free volume indicated by Tg reduction (Table 2). For example, a significant increase for polymers EL, RS, and IR are accompanied by a large Tg decrease of 8.3, 5.3, and 5.3 °C, respectively. However, this is not a general trend since RL with a large Tg drop exhibits only a small (0.1%) ER change (Table 6).

**Table 6.** Results of work of compaction (Wc), elastic recovery (ER%), yield pressure parameters (Heckel $P_Y$, Kawakita 1/b, slope S' of derivatized (dx/dF vs. 1/F') compression curves and tensile strength (TS) (mean values ± SD, $n$ = 3 and LSD$_{0.025}$).

| Polymer | Wc (J) | ER (%) | $P_Y$ (MPa) | 1/b (MPa) | S' (mm) | TS (MPa) |
|---|---|---|---|---|---|---|
| EL [a] | 4.26 ± 0.24 | 7.78 ± 0.44 | 123.8 ± 3.6 | 57.0 ± 1.9 | 1.68 ± 0.02 | 3.71 ± 0.11 |
| ELex [a] | 2.68 ± 0.13 | 9.64 ± 0.15 | 105.8 ± 1.7 | 37.0 ± 1.6 | 0.83 ± 0.02 | 2.68 ± 0.22 |
| EE | 2.17 ± 0.03 | 13.36 ± 0.08 | 41.6 ± 1.1 | 63.4 ± 1.3 | 1.19 ± 0.02 | 2.79 ± 0.13 |
| EEex | 2.04 ± 0.16 | 13.80 ± 0.65 | 168.1 ± 1.4 | 77.8 ± 1.4 | 1.11 ± 0.01 | 2.54 ± 0.11 |
| RS [a] | 2.86 ± 0.03 | 8.72 ± 0.29 | 85.2 ± 2.2 | 39.3 ± 0.2 | 1.38 ± 0.08 | 2.72 ± 0.24 |
| RSex [a] | 2.44 ± 0.81 | 10.19 ± 0.95 | 73.0 ± 1.7 | 30.1 ± 0.6 | 0.93 ± 0.01 | 1.23 ± 0.21 |
| RL | 2.10 ± 0.08 | 13.22 ± 0.48 | 74.0 ± 2.2 | 88.0 ± 0.3 | 1.02 ± 0.02 | 1.97 ± 0.06 |
| RLex | 2.21 ± 0.33 | 13.32 ± 0.68 | 122.4 ± 2.6 | 95.9 ± 1.8 | 0.95 ± 0.01 | 0.68 ± 0.02 |
| VA | 3.32 ± 0.14 | 11.10 ± 0.23 | 52.1 ± 1.4 | 27.3 ± 0.5 | 1.54 ± 0.05 | 9.20 ± 0.16 |
| VAex | 2.24 ± 0.12 | 11.83 ± 0.42 | 90.1 ± 3.7 | 82.1 ± 0.9 | 0.93 ± 0.02 | 3.63 ± 0.11 |
| IR | 1.43 ± 0.13 | 17.02 ± 0.9 | 86.4 ± 3.7 | 15.9 ± 0.5 | 1.65 ± 0.01 | 0.81 ± 0.20 |
| IRex | 0.55 ± 0.17 | 20.82 ± 0.72 | 90.1 ± 2.6 | 43.2± 2.5 | 0.84 ± 0.01 | 0.39 ± 0.04 |
| SR [a] | 4.18 ± 0.36 | 6.89 ± 0.21 | 70.3 ± 1.2 | 25.3 ± 0.1 | 1.87 ± 0.13 | 12.31 ± 0.34 |
| SRex [a] | 3.09 ± 0.12 | 7.01 ± 0.72 | 71.9 ± 1.7 | 39.3 ± 0.8 | 1.09 ± 0.15 | 6.74 ± 0.51 |
| SL [a] | 2.94 ± 0.08 | 8.89 ± 0.24 | 70.01 ± 0.9 | 35.4 ± 1.3 | 1.35 ± 0.02 | 2.91 ± 0.06 |
| SLex [a] | 2.88 ± 0.07 | 9.67 ± 0.56 | 72.1 ± 2.0 | 37.7 ± 0.9 | 1.23 ± 0.06 | 2.56 ± 0.09 |
| LSD$_{0.05}$ | 0.11 | 0.36 | 0.95 | 2.06 | 0.02 | 0.08 |

[a] Values reprinted from https://doi.org/10.1016/j.cherd.2020.01.035.

### 3.7.2. Compression Model Parameters

Yield pressure parameters as indices of plastic deformation were obtained from Heckel ($P_Y$) and Kawakita (1/b) models representing two different approaches of compression data analysis [30,32]. Typical profiles for polymer EE from the methacrylic and VA from the polyvinyl chemical groups are presented in Figure 5. Computed $P_Y$, 1/b values are presented in Table 6. Except for EL and RS, the other polymers showed an increase in $P_Y$ and 1/b of the extrudate powders implying reduced plasticity. Conversely, ELex and RSex extrudate powders showed a decrease (Table 6) implying softening. The effect of HME on $P_Y$ and 1/b was statistically significant (differences > LSD$_{0.05}$ = 0.95 MPa and >LSD$_{0.05}$ = 2.06 MPa, respectively) and was greater for polymers EE, RL VA, IR, and smaller for SR and SL (Table 6). These results are in agreement with previous findings [16,17,19,43].

In addition to the above well-known compression models, the slope (S') of derivative compression curves (dx/dF vs. 1/F') was estimated as an index of deformability. High S' value indicates predominantly plastic deformation and low brittleness [27]. Representative plots are shown in Figure 5c and it is seen that the points more or less fall on straight lines. S' values are given in Table 6. In all cases, the decrease due to HME was statistically significant (differences > LSD$_{0.05}$ = 0.06). For EL, RS, VA, IR, and SR the decrease was greater. For 6 out of 8 polymers the estimations from S' are in agreement with the Heckel and Kawakita models, but for EL and RS there is disagreement since the extrudate powder had lower S' (Table 6) implying loss of plasticity whereas $P_Y$ and 1/b were lower indicating the opposite (Table 6). ANOVA (Table 4) showed a significant interaction of polymer and HME for all three deformability parameters.

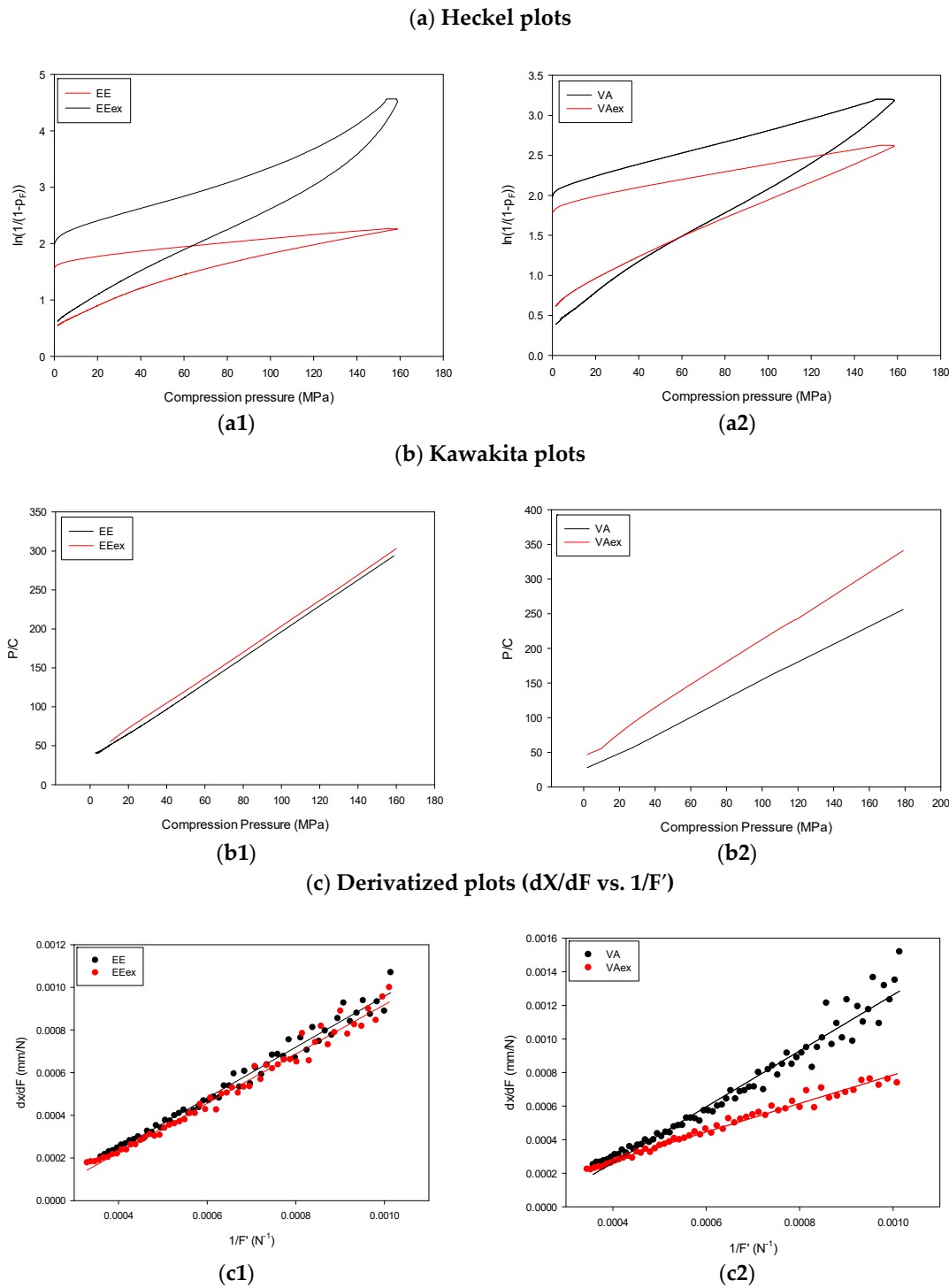

**Figure 5.** Plots of Heckel (**a**), Kawakita (**b**), and derivatized compression curve (**c**) for neat/extruded Eudragit EPO (**a1,b1,c1**) and Kollicoat VA 64 (**a2,b2,c2**) powders (neat in black, extruded in red color).

### 3.7.3. Tensile Strength of Tablets

Tensile strength (TS) is an important attribute for tablet quality [54] evolving from inter-particle bonds. The results are presented in Table 6. A statistically significant decrease due to HME is observed in all cases (differences > $LSD_{0.05}$ = 0.08 MPa). This is mainly due to the higher elastic recovery of the extrudate powders and reduced plastic deformation. Polymers EE and SL were less affected by HME showing small TS reduction (8.9% and 12.0%) whereas RS, RL, VA, IR, and SR were more affected

showing great reduction (54.8%, 65.5%, 60.55%, 51.85%, and 45.25%). More importantly, considering 1 MPa as the threshold for acceptable TS, it is seen in Table 6 that from the more affected polymers, VA and SR are still able to form strong tablets (strength 3.63 and 6.74 MPa), but RL and IR form weak tablets (strength 0.68 and 0.39 MPa) that may not be able to process in high-speed industrial machines. Overall, HME incapacitated tablet formation for 2 out of 8 common pharmaceutical polymers. ANOVA (Table 3) showed a significant interaction with the effects of polymer and HME treatment.

### 3.8. Statistical Processing

#### 3.8.1. Pearson's Correlation Coefficient

Pearson's coefficient ($r$) was used as an index for the early identification of correlations between variables. Its value ranges from −1 (negative correlation) to 1 (positive) with zero indicating the absence of linear correlation. In Figure 6 correlation coefficients are presented as a heat map. Reading downwards along the rows it is seen that densities $\rho_t$ and $\rho_b$ are positively correlated ($r = 0.889$). HR and CC correlate negatively with d50 ($r = -0.499$, $r = -0.524$) due to the easier packing of larger particles and positively correlated with each other ($r = 0.983$) as they express the same powder property. The surface free energy ($\gamma_s$) is negatively correlated with $\rho_t$ ($r = -0.492$) which is due to the effect of $\gamma_s$ on particle–particle interactions and packing [55].

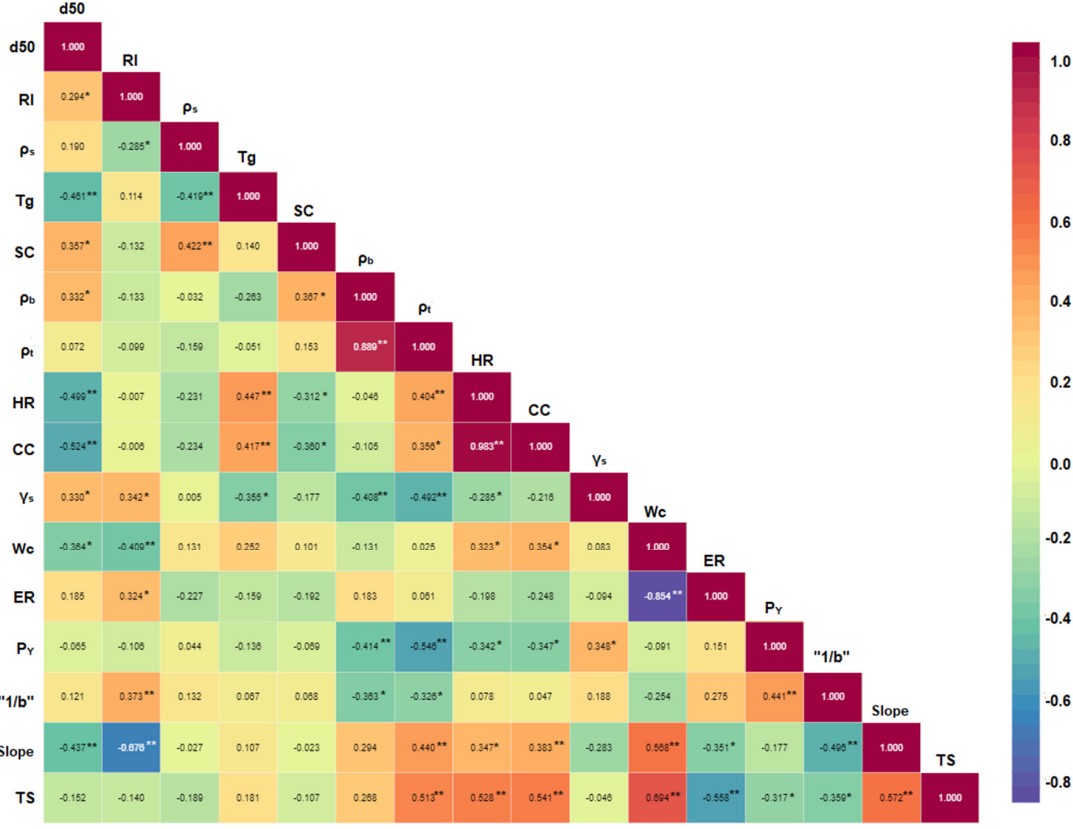

**Figure 6.** Heat map of Pearson's correlation coefficient between the studied variables (* Correlation significant at $p = 0.05$; ** Correlation significant at $p = 0.001$).

Turning to compression, ER shows a negative correlation with Wc ($r = -0.854$). This is because ER is proportional to decompression work represented with area OF′X′ in Figure 1 which is subtracted from the total force/displacement area (XF′X′) to give Wc. From the deformability indices, the slope of derivative compression curves S′ is positively correlated with Wc ($r = 0.568$) implying that they express a common compaction mechanism. This should be related to plastic deformation since S′ is

negatively correlated with Kawakita yield pressure 1/b ($r = -0.501$). As expected, tensile strength (TS) is positively correlated with Wc ($r = 0.694$), but negatively with ER ($r = -0.558$) [56]. It is interesting that TS shows better correlation with S' ($r = 0.572$) than the yield pressure parameters $P_Y$ ($r = -0.317$) and 1/b ($r = -0.359$). The positive correlation of TS with packing parameters $\rho_t$, HR, and CC should be attributed to differences between neat and extrudate powders reflected on these indices.

### 3.8.2. Principal Component Analysis (PCA)

PCA with VARIMAX rotation (Hair et al., 2010) was applied to neat and extrudate polymer data to compress the variables into principal components and identify relationships. The following variables were used: particle diameter (d50), glass transition temperature (Tg), tap density ($\rho_t$), surface free energy ($\gamma_s$), work of compaction (Wc), elastic recovery (ER), Heckel ($P_Y$) and Kawakita (1/b) yield pressures, the slope of derivative compression curve (S'), and tensile strength (TS). Shape index and electrostatics were not included in the PCA because they did not add further information. (repetition)

Results of PCA are presented in Figure 7 as a scree plot of Eigenvalues against the component number and in Table 7 as components with loadings > 0.50 (absolute values). Eigenvalue > 1 was used to identify significant components. From Figure 7 it appears that Eigenvalue is dropping with the component number and after the third elbow it becomes less than one. Therefore, the three components were considered significant. Parallel Analysis [57] confirmed the statistical significance of these components as the corresponding Eigenvalues were greater than the critical ones estimated by Parallel Analysis. In addition, all the communalities (Table 7) are greater than 0.55 (average communality 0.72) which confirms the adequacy of the PCA model using three components. They explained 71.45% of the total variance and their loadings are presented in Table 7. The first component represents compaction. It consists of Wc, S' and TS with high positive loadings (0.944, 0.575, and 0.746) contributing to the compaction process and ER with negative loading ($-0.883$) opposing compaction. The second component represents rigidity and interactivity. It consists of $\gamma_s$, $P_Y$, and 1/b with positive loadings (0.588, 0.755, and 0.529) contributing to rigidity and interactivity opposing close packing expressed by $\rho_t$ with negative loading ($-0.896$). The third component contains d50 with positive (0.814) and Tg with negative loading ($-0.801$) which is interpreted by the tendency of neat powders with higher Tg to have smaller particle diameters (Table 2).

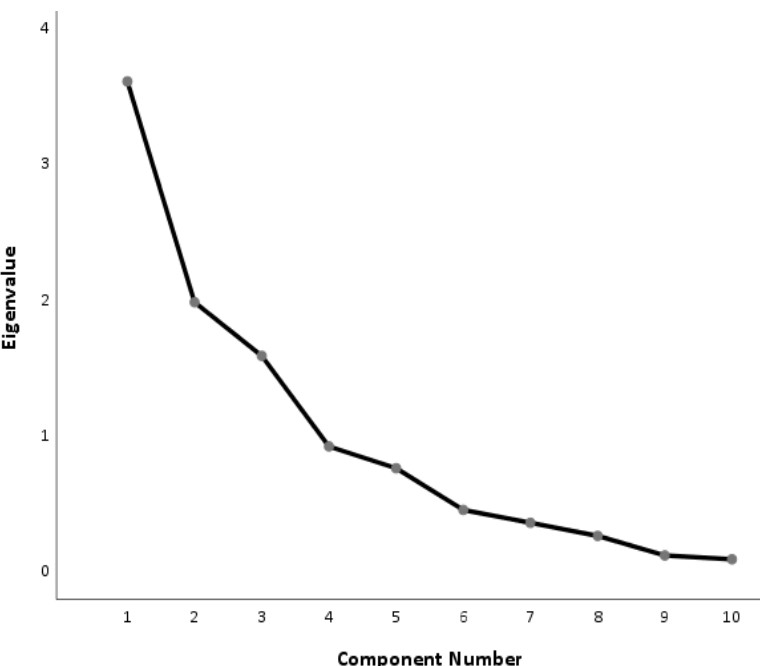

**Figure 7.** Scree plot of Eigenvalue against component number obtained from PCA.

**Table 7.** Components obtained from the Principal Component Analysis (varimax rotation) with loadings greater than 0.5.

| Variables | Component 1 | Component 2 | Component 3 | Communalities |
|---|---|---|---|---|
| Mean particle diameter (d50) | | | 0.814 | 0.733 |
| Glass transition temperature (Tg) | | | −0.801 | 0.651 |
| Tapped density ($\rho_t$) | | −0.896 | | 0.803 |
| Surface free energy ($\gamma_s$) | | 0.588 | | 0.793 |
| Work of compaction (Wc) | 0.944 | | | 0.930 |
| Elastic recovery (ER) | −0.883 | | | 0.783 |
| Heckel's yield pressure ($P_Y$) | | 0.755 | | 0.576 |
| Kawakita parameter (1/b) | | 0.592 | | 0.593 |
| Slope (S') | 0.575 | | | 0.634 |
| Tensile strength (TS) | 0.746 | | | 0.748 |
| *Parameters of the analysis* | | | | |
| Eigenvalue | 3.60 | 1.97 | 1.58 | |
| Critical Eigenvalues (Parallel Analysis) | 2.04 | 1.67 | 1.45 | |
| Variance explained (%) | 36.00 | 19.70 | 15.75 | |
| Cumulative explained variance (%) | 36.00 | 55.70 | 71.45 | |

Kaiser–Meyer–Olkin's measure of sampling adequacy: 0.591—Bartlett's test of sphericity: $p < 0.001$, $\chi^2 = 263.8$, $df = 45$.

### 3.8.3. Hierarchical Cluster Analysis (HCA)

HCA was used to classify the polymers based on particulate, powder, and compression properties regardless of HME treatment using the polymers' scores on the three components derived from PCA. This decision was taken to avoid the problem of multicollinearity among the original variables. The results are depicted in Figure 8 as a dendrogram where the X-axis represents the rescaled distance Cluster Combine Index (CCI). The lower the index, the greater the dissimilarity with other clusters. It is seen that the sixteen cases (8 polymers × 2 treatments) are assembled into two major clusters A and J. Cluster A is composed of polymers EL, RLex, and EEex. It has CCI 12 and therefore its dissimilarity to the others is low. Cluster J consists of several sub-clusters of which B has only polymer VA that can be seen as an outlier, C has polymers VAex and RL and D has only polymer EE. Cluster G is comprised of two strong clusters E and F with CCI 5, and therefore it is dissimilar to the others described above. Of these, cluster E contains polymers ELex, IR, and RS whereas F consists of two sub-clusters: F1 which contains polymer IRex and F2 polymers RSex, SRex, SR, SLx, and SL. As F2 has a very low CCI value of 2 it is a very strong cluster very dissimilar from the others. Therefore, the polymers RSex, SRex, SR, SLx, and SL that belong to F2 are interchangeable regarding their solid-state properties. Additionally, from these, polymers RSex and SRex having the same pharmaceutical functionality as sustain release matrix formers could be used to substitute each other should the need arises. In broad terms, it could be said that the IRex of cluster E is also similar to the polymers in F1 since both F1 and F2 are under cluster F with CCI 5 and this could give one more exchangeable pair of IRx and SLx being both solubility improvers.

As dissimilarity is based on CCI value ≤ 5 clusters F and E with index 3 and 4 respectively are of particular interest. Cluster F is formed by subcluster F1 containing polymers SR and SL both as neat (SR, SL) and extrudates (SRex, SLex), and by subcluster F2 containing IRex. Although the above polymers belonging to the same cluster F with CCI ≤ 5 are exchangeable this is practically useful only for SLex, IRex that have the same functionality (dissolution aid). Cluster E contains polymers RS, IR, and ELex with different functionalities. However, the combination of clusters E and F gives cluster G with CCI5, indicating dissimilarity to other clusters, or looking at it from a different angle, polymers in G have closely related solid-state properties but different from polymers in other clusters. Hence, they could be exchangeable in terms of particulate/powder/compaction properties. In particular, RS can be used as a replacement for SR (sustain release) and IR for SL (dissolution aid).

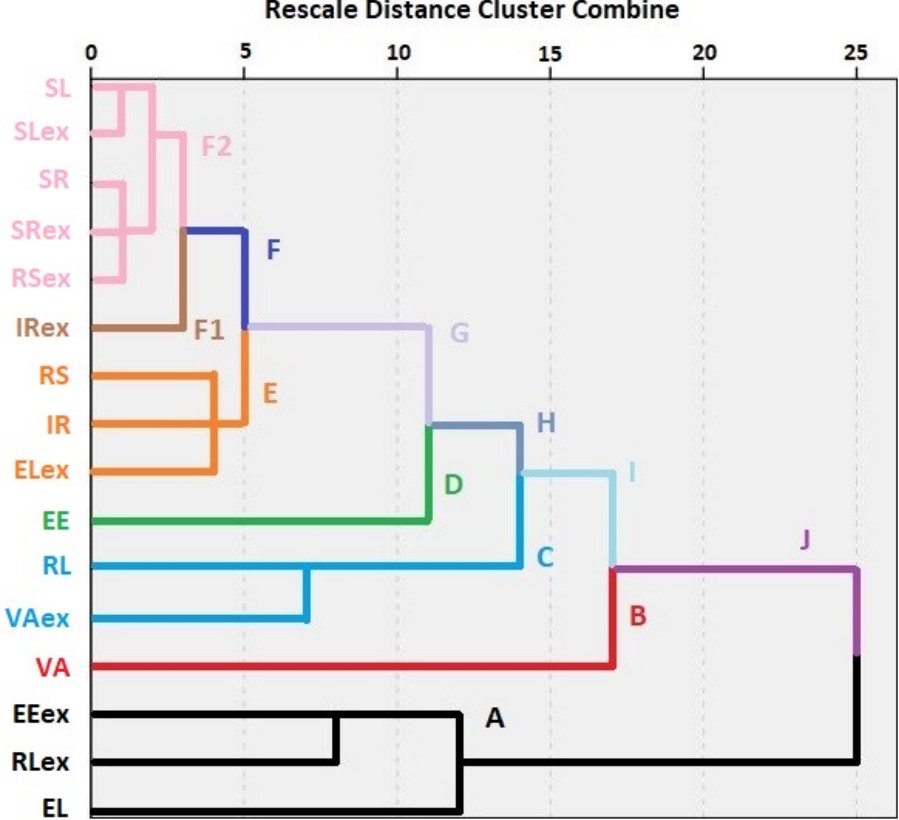

**Figure 8.** Dendrogram of polymer against the distance cluster combine index showing results of the Hierarchical Cluster Analysis (HCA).

## 4. Conclusions

This study explored the impact of HME on particulate, powder, and compaction properties of 16 neat and extruded pharmaceutical polymers. The effects of electrostatics induced by the HME on the packing indices were demonstrated and the contribution mechanism of the slope S' on the derivative force/displacement curve on the mechanical strength of tablets was confirmed. The principal component analysis was able to distinguish variables having similar or dissimilar effects for a certain process, and hierarchical cluster analysis successfully clustered the polymers based on solid-state properties regardless of HME treatment. HCA can be useful to select exchangeable polymers without significant differences in their solid-state properties and with the same functionality, e.g., sustain release or dissolution enhancement if the need arises, for example, due to shortage, financial cost, or incompatibility. For future research, it will be interesting to apply different analysis strategies to determine significant principal components that will enter HCA analysis and see how that may affect the result of the clustering process. Also, it will be interesting to further explore the PCA/HCA method in conjunction with literature sources and further experimental data to find the exchangeability of excipients in other pharmaceutical applications.

**Author Contributions:** Conceptualization, I.P. and I.N.; methodology, I.P. and I.N.; software, I.P., G.M., and I.N.; investigation, I.P., M.T., and F.-M.S.; resources, I.P. and I.N.; data curation, I.P., G.M., and I.N.; writing—original draft preparation, I.N. and I.P.; writing—review and editing, I.N., I.P., and G.M.; visualization, I.P. and I.N.; supervision, I.P. and I.N.; project administration, I.N. All authors have read and agreed to the published version of the manuscript.

**Funding:** This research received no external funding.

**Acknowledgments:** We are very grateful to the reviewers for their contribution.

**Conflicts of Interest:** The authors declare no conflict of interest.

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
