# Peer review of "Impact of Hot-Melt-Extrusion on Solid-State Properties of Pharmaceutical Polymers and Classification Using Hierarchical Cluster Analysis"

_processes, doi:10.3390/pr8101208_

Round 1

Reviewer 1 Report

Review from Manuscript ID: processes-945767

Summary: The paper describes the impact of hot melt extrusion on solid state properties of  polymers used in the pharmaceutical industry.

The paper is well written and structured. The experiments are well conducted and support the conclusions. I only have a few comments and remarks that I would like to be clarified.

  • Line 179: “In a PCA plot…” there are several plots that can be drawn from a PCA. Additionally, scores and loadings have different meanings. Scores are the points correspondent to the samples (lines in the data matrix), in this case the polymers. Loadings are related with the variables (columns in the data matrix), in this case the slid state properties. Also, the explanation of the colocation of the points in the score or loading plot is a bit redundant. The concepts regarding PCA need to be clarified and rewritten. It is also necessary to write the type of data pre-processing used before the PCA.
  • Line 448: Why choose an eigenvalue > 1 to determine the number of principal components? There are several strategies to determine the #PC, cross-validation, for example.
  • Line 482: why chose a CCI < 5?
  • Line 509: It is HCA and not HCE.

Author Response

Manuscript ID: processes-945767

 Please find below replies to the comments in the order as received.

In the revised manuscript all additions are in red color.

Comment #1 The paper describes the impact of hot melt extrusion on solid state properties of polymers used in the pharmaceutical industry. The paper is well written and structured. The experiments are well conducted and support the conclusions.

Response Thank you very much for the positive comment.

Comment #2 Line 179: “In a PCA plot…” there are several plots that can be drawn from a PCA. Additionally, scores and loadings have different meanings. Scores are the points correspondent to the samples (lines in the data matrix), in this case the polymers. Loadings are related with the variables (columns in the data matrix), in this case the solid state properties.

Response Text has been changed in accordance with the reviewer’s comment. All methodological descriptions for the PCA and HCA were reduced only to the necessary ones (lines 185-188 of revised).

Comment #3 Also, the explanation of the coloration of the points in the score or loading plot is a bit redundant.

Response Figure 8 was removed since it does not add information further to Components and loadings Table 7.

Comment #4 The concepts regarding PCA need to be clarified and rewritten.

Response Methodological descriptions for PCA were clarified and sections 2.2.11 and 3.8.2 rewritten.

Comment #5 It is also necessary to write the type of data pre-processing used before the PCA.

Response - PCA was performed using as “input” the matrix of pair-wise Pearson’s correlations among the variables. Variables’ values were automatically transformed (by the software) to z-scores. Prior to PCA, data were checked for outliers. The linear relationships among the tested variables were examined by inspecting the corresponding scatter plots. The quality of the data was confirmed from the satisfactory value of KMO index of sampling adequacy (≈0.60>0.50) and the statistical significance of the Bartlett’s test of sphericity (p<0.001). Text has been added in the revised, lines 190-194.

 Comment #6 Line 448: Why choose an eigenvalue > 1 to determine the number of principal components? There are several strategies to determine the #PC, cross-validation, for example.

Response -  Parallel Analysis (reference by Horn 1965 has been added in the revised) confirmed the statistical significance of the three components, since the corresponding observed eigenvalues were greater than the critical ones obtained by the analysis. In addition, all the communalities (Table 7) were greater than 0.55 (average communality 0.72) which confirm adequacy of the PCA model using the three components. Text has been added in the revised lines 478-483. In addition, an extra row has been added in Table 7 with the critical eigenvalues estimated by the Parallel Analysis.

Comment #7 Line 482: why chose a CCI < 5?

Response - Since this is a controversial issue CCI = 5 was removed as a criterion for dissimilarity.

Comment #8 Line 509: It is HCA and not HCE.

Response - It has been corrected.

Reviewer 2 Report

The authors are to be congratulated on producing a quality manuscript. There are a few minor issues that they may wish to address. First, the implementation of hot-melt extrusion to process polymers in powder to tablets will minimize stability risks and production time. However, HME is likely going to affect negatively or positively the strain tolerance under compression and this may promote cracking distress to some extent. The authors may reference the appropriate literature to place their findings in context. Also, the decrease in glass transition temperature through the addition of various plasticizers can lead to polymers of high resistance against cracking which the authors may wish to briefly discuss. Finally, the authors could reference the standards and norms utilized to perform the experiments. The impact of cyclic compression load worth to be evaluated in a future research.

Author Response

Manuscript ID: processes-945767

Please find below replies to the comment in the order as received. In the revised all additions are in red color.

Comment #1 The authors are to be congratulated on producing a quality manuscript.

Response - Thank you very much

Comment #2 There are a few minor issues that they may wish to address. First, the implementation of hot-melt extrusion to process polymers in powder to tablets will minimize stability risks and production time.

Response - We agree

Comment #3 However, HME is likely going to affect negatively or positively the strain tolerance under compression and this may promote cracking distress to some extent. The authors may reference the appropriate literature to place their findings in context.

Response – We agree. Text has been added in the revised (lines 52, 53) addressing this point and a new reference [18] has been added.

Comment #2 Also, the decrease in glass transition temperature through the addition of various plasticizers can lead to polymers of high resistance against cracking which the authors may wish to briefly discuss.

Response – We agree. Text has been added in the revised (lines 53-55) together with supporting reference [3].

Comment #3 ‘… the authors could reference the standards and norms utilized to perform the experiments.

Response - United States Pharmacopoeia monograph 1217 (USP35, 2012) has been referenced in the revised, line 169 of as the standard for conducting tests of the mechanical strength of tablets. Also reference [53] has been added.

Comment #4 The impact of cyclic compression load worth to be evaluated in a future research.

Response - We do agree

Reviewer 3 Report

This manuscript provides a study of the physical changes of polymer used in Hot-Melt-Extrusion for pharmaceutical applications. The article is interesting and is developed under scientific standards. I would like to support its publication after some comments are taken into consideration:

My major concern is related with the discussion of the obtained results. The authors have analyzed the physical properties of polymers but its relation with the effects produced during hot melt extrusion process is not clear and very few conclusions are obtained.

Other minor considerations:

  • The samples tagging hinders the comprehension of the manuscript. In some parts it becomes very difficult to follow the ideas exposed in the article.
  • Section 3.1 (Hot-melt extrusion) is a bit confusing. What is the main idea of the section? There is a relation between barrel pressure, temperature extrusion and Tdec?
  • In section 3.2 there is no analysis of the SEM microscopy images. In my opinion I think that particle morphology is well observed in the images which may contribute to the manuscript conclusions.
  • Observing the DSC profile (figure 3) the claim that: “all polymers are amorphous except IR which is semi-cristalline” seems inaccurate.
  • The differences between electrostatic surface charge values is significative? Its not clear that changes of 1-2 nC may be attributed to HME.   
  • Considering the amount of work that contains this article I think that the conclusion parts is short and should be improved.

Author Response

Manuscript ID: processes-945767

Please find below replies to the comments in the order as received. In the revised manuscript all additions are in red color.

Comment #1 This manuscript provides a study of the physical changes of polymer used in Hot-Melt-Extrusion for pharmaceutical applications. The article is interesting and is developed under scientific standards. I would like to support its publication after some comments are taken into consideration

Response Thank you very much

Comment #2 My major concern is related with the discussion of the obtained results. The authors have analyzed the physical properties of polymers but its relation with the effects produced during hot melt extrusion process is not clear and very few conclusions are obtained.

Response – The main idea of the work was to show distinct effects of HME on the polymer properties and then to cluster the polymers regardless of chemistry and HME treatment. This is why there is no in-depth discussion of the nature of the HME effects. However, the reviewer’s suggestions have been taken into consideration in the revised manuscript and text has been added in appropriate points (lines 237-240, 268-270, 289-291, 318) relating the HME effects with the shearing and heating occurring during the process.

Other minor considerations (MC):

MC#1 The samples tagging hinders the comprehension of the manuscript. In some parts it becomes very difficult to follow the ideas exposed in the article.

Response - Because of the large number of experimental polymers it was unpractical using full commercial names, abbreviations as short as possible were used instead. However, the presentation of the work has been thoroughly revised to ease reading, and the following changes have been made in the manuscript:

- Codes of polymer samples are given early in the manuscript in Table 1 and the Table has been simplified. Subscript for extrudate powders is always ex (i.e. Polymerex) in the text and in the Tables.

- In Figure 2 full names of the samples are given in the legends.

MC#2 Section 3.1 (Hot-melt extrusion) is a bit confusing. What is the main idea of the section? There is a relation between barrel pressure, temperature extrusion and Tdec?

Response - The idea in this section is to provide information on the behavior of the polymers during HME process which will help the reader going through the rest of the manuscript.

MC#3 In section 3.2 there is no analysis of the SEM microscopy images. In my opinion I think that particle morphology is well observed in the images which may contribute to the manuscript conclusions.

Response - Sentence has been added in revised, line 237-240. Furthermore, since the aim of the work was to compare and classify polymers on the basis of a range of solid state properties, these properties are not mentioned individually in the conclusions.

 MC#4 Observing the DSC profile (figure 3) the claim that: “all polymers are amorphous except IR which is semi-cristalline” seems inaccurate.

Response - In the insert of Figure 3 the DSCs of Kollicoat IR neat and extruded forms show a step-type glass transition between 40 and 50 oC and a crystalline phase melting between 205 oC and 215 oC. This is what we describe as semi-crystalline. Other polymers did not show melting endotherm.

MC#5 The differences between electrostatic surface charge values is significative? Its not clear that changes of 1-2 nC may be attributed to HME.

Response - The measured values are small because no tribeletrification was applied before the experiments, i.e. measurements were taken on equilibrated powders. Text has been added in the revised (lines 275-277) to point out this condition. However, they measurement showed clear, significantly different electrostatic surface charges.

MC#6 Considering the amount of work that contains this article I think that the conclusion parts is short and should be improved.

Response - Conclusions have been expanded by adding text in the end, lines 543-547.